# Determination of some heavy metals and their health risk in T-shirts printed for a special program

**Milkessa Fanta Sima** [ID]*

Department of Chemistry, College of Natural and Computational Science, Mattu University, Mattu, Ethiopia

* milkikoo@gmail.com, milkessa.fanta@meu.edu.et

**Data Availability Statement:** All relevant data are within the paper.

**Funding:** The authors received no specific funding for this work.

## Abstract

Heavy metals often are used in different textile processes, like dyeing and printing. When the toxic elements are present in more than recommended in textile materials they may impose potential risk to human health by absorption through the skin. In this study concentrations of some heavy metals (Co, Cu, Cr, Cd, and Pb) were analyzed in skin contact fading T-shirts printed for a special program at Mettu town using Atomic Absorption Spectroscopy with a microwave digestion method technique for sample preparation. High levels of Cu were found in black, green, blue, and red-colored T-shirts ranging from 26.726–179.315mg/kg. Cr exceeded the recommended limits in most samples of T-shirts and was mostly in yellow, black, and blue colors. Cd levels were found to be within normal ranges. However, all T-shirt samples had low levels of cobalt, ranging from 1.33±2.13 to3.94±0.21. Maximum lead concentrations were found to be 3.40 ± 0.19 mg/kg for red-colored samples and 2.71 ± 0.13 mg/kg for blue colored samples. The metal concentrations in the T-shirts investigated were also compared to the OEKO Tex standard 100 limits. In this investigation, the concentrations of Pb, Cu, and Cr in red and green colored T-shirt samples were above the OEKO Tex suggested standard value. Therefore a strict local and international regulation and measures need to be taken to avoid toxicity of the studied metals.

## Introduction

Quality of textile production is very important because people want to be able to buy clothing, bedding, and household textiles that have been tested and are not dyed in any way with harmful substances [1–3]. People are constantly in contact with textiles through wearable clothing, cleansing apparel, carpeting, furniture cover, and bedding; due to this reason, we are often exposed to different allergenic and toxic chemicals coming from these textiles via inhalation, ingestion, and dermal absorption. The textile can be made of natural or synthetic fibers and the whole production process involves extensive use of chemicals. The main chemical pollutants present in textiles are dyes containing carcinogenic amines, metals, pentachlorophenol, chlorine bleaching, halogen carriers, free formaldehyde, biocides, fire retardants, and softeners [4, 5].

**Competing interests:** The authors have declared that no competing interests exist.

Heavy metals often are used in different textile processes, like dyeing and printing [6]. Toxic and allergic metals including Co, Cu, Cr, and Pb are used as metal complex dyes, Cr as pigments mordant, Sn as a catalyst in synthetic fabrics and as synergists of flame retardants, Ag as antimicrobials, and Ti and Zn as water repellents and odor preventive agents [7–10]. Raw textile materials may also contain heavy metals [11]. Cotton, flax, and hemp sometimes adsorb very large amounts of metals from the environment [12, 13] and can be used as bio-absorbers [14].

When toxic elements are present in high amounts in textile materials they may impose potential risk to human health by absorption through the skin including skin alterations i.e. dermatitis, irritation, allergy, and skin micro-flora reduction [15]. It is well known that some metals, such as cobalt, chromium, copper, and nickel, are skin sensitizers[16, 17] while other trace elements (e.g. cadmium and lead) are highly toxic and carcinogenic[18] Cr can lead to liver damage, pulmonary congestion, and cancer [19]. Lead causes neurotoxin. It affects the human brain and cognitive development, as well as the reproductive system. Chromium (VI), (Hexavalent Chromium) is recognized as a human carcinogen and is linked to lung, respiratory system, and sinus cancers. Whereas copper imbalance causes arthritis, fatigue, insomnia, migraine headaches, depression, panic attacks, and attention deficit disorder. The purpose of this study was to determine the concentrations of some heavy metals (Pb, Cd, Cr, Co and Cu) in skin-contact fading T-shirts printed for fun and special ceremonies selected randomly from the printing house and markets of Mettu town by Flame Atomic Absorption Spectroscopy and know the pollution status of the studied materials.

## Materials and methods

### Sample preparation

T-shirt samples in various colors, including black, red, blue, yellow, and green, were purchased at random from Mettu town printing houses and super markets. After drying 0.5 g of each sample in $HNO_3$, it was digested in a Microwave Digestion System (Titan microwave sample preparation system, India) for 5 minutes at 105˚C, then 15 minutes at 180˚C, and finally 20 minutes at 200˚C to determine heavy metal content of T-shirt materials. Extracts were filtered and brought to a volume of 25 mL with ultrapure water after chilling, and AAS was used to analyze them. Blanks made with the same amount of reagents but without the sample were made under the same conditions. The accuracy of the instrumental procedures was tested using blank and control samples, as well as reference materials. A plot of standard metal concentration versus matching absorbance was made for each tested metal, and the calibration curves' correlation coefficient ($R^2$) and regression equations were recorded. For the samples and standard solutions, all measurements were done in triplicate, and the mean results were used, along with the appropriate standard deviation values, for statistical analysis.

### Elemental quantitative analysis

The analysis was carried out using an atomic absorption spectrophotometer (Buck Scientific model 210 VGP) with a deuterium arc background corrector and a standard air-acetylene flame setup. For maximum signal intensity, the instrument's operating settings were optimized. At the corresponding primary source line, a hollow cathode lamp for each metal (Co, Cu, Cr, Cd, and Pb) was utilized, which was operated according to the manufacturer's recommendations. To guarantee acceptable flame conditions, the acetylene and air flow rates were regulated. To determine the elements in the digested blank solutions, the same analytical approach was used. Table 1 summarizes the optimized operational radiation wavelength, slit

**Table 1. Instrumental operating conditions for the determination of Co, Cu, Cr, Cd, and Pb using AAS.**

| Element | Parameters | | | |
|---|---|---|---|---|
| | Wavelength/nm | Slit/nm | Lamp current, mA | Instrument detection limit |
| Pb | 283.3 | 1.0 | 5 | 0.04 |
| Cd | 228.9 | 0.5 | 4 | 0.01 |
| Cr | 357.9 | 0.2 | 7 | 0.04 |
| Co | 240.7 | 0.2 | 3 | 0.05 |
| Cu | 324.7 | 0.5 | 5 | 0.005 |
| Gas mixture | Acetylene/air | | | |

width, and lamp current employed in the AAS measurements, as well as the detection limit for each analyzed heavy metal in the examined T-shirt samples.

## Method validation

The accuracy of digestion procedure and efficiency of the AAS instrument were checked by spiking sample with known concentration of the analyte. Spiked samples were prepared by adding a small known quantity of metal standard solutions to T-shirt samples by applying similar digestion procedure and analyzing for the levels of metals and calculating the recovery percent. Percent recovery (R) was calculated [20] using the Eq (1):

$$\frac{C_s - C}{S} * 100 \tag{1}$$

Where, Cs = metal concentration of the spiked sample

C = metal concentration of the non-spiked sample

S = concentration equivalent of analyte added to the sample.

## Results

Calibration curves were prepared to determine the concentration of metals in the samples solution. A series of standard working solution were prepared from their respective salt. The concentration range, the correlation coefficients and the correlation equations of the calibration curves for the determination of metals in the samples by AAS are given in Table 2. The correlation coefficients of all the calibration curves were > 0.999 and these correlation coefficients showed that there was very good correlation (relationship) between concentration and absorbance.

## Method detection limit and limit of quantification

In this study Limits of detection (LOD) and limit of quantification (LOQ) were calculated from three and ten times the standard deviations for blank measurements divided by the

**Table 2. Concentrations of working standard solutions, correlation coefficients and regression equations of the calibration curves for the studied metals.**

| Analyzed heavy metals | Concentration range (mg/kg) | Regression equation | $R^2$ |
|---|---|---|---|
| Cu | 25, 50, 75, 100, 125, 150, 175, 200 | y = 0.0085x- 0.0089 | 0.9941 |
| Cr | 0, 1, 2, 3, 4, 5, 6, 7 | y = 0.3002x- 0.0008 | 0.9975 |
| Co | 0, 0.5, 1.0, 1.5, 2.0, 2.5, 3.0 | y = 0.1675x+0.0086 | 0.997 |
| Cd | 0, 0.02, 0.04, 0.06, 0.08, 0.1, 0.2, 0.3 | y = 0.467x+0.0023 | 0.9977 |
| Pb | 0, 0.5, 1.0, 1.5, 2.0, 2.5, 3.0 | y = 0.3989+0.0022 | 0.9954 |

**Table 3. Method detection limit and limit of quantification.**

| Metals | Method detection limit | Limit of quantification |
|---|---|---|
| Cu | 12.99 | 39.36 |
| Cr | 0.283 | 0.859 |
| Co | 0.145 | 0.441 |
| Cd | 0.011 | 0.032 |
| Pb | 0.051 | 0.399 |

calibration curve slope, respectively. Each blank solution was run with AAS for the metals level in similar manner as the samples and the values obtained for all metals are listed in Table 3.

## Accuracy

In this study, the method validation was made by the spiking experiment in which known quantities of the metals standard solution were added to the samples and applied the whole procedure to the mixture (spiked samples) and calculated the percent recoveries.

The obtained percentage recovery varied from 95.33% to 101.66% which were in the acceptable range (Table 4).

## Analysis of metals in T-shirts

In this study average concentrations of heavy metals (Co, Cu, Cr, Cd, and Pb) were analyzed using AAS in green, blue, black, yellow, and red colored T-shirts whose results are presented in Table 5.

**Copper**: high concentration of copper was found in blue, green, black and red color t-shirt and it is about 177.01± 2.305 mg/kg, 127.16± 6.741 mg/kg, 93.37 ± 3.192 mg/kg and 93.37 ± 3.192 mg/kg respectively.

**Chromium**: The study show concentration of chromium in yellow, black, blue, red and green color t-shirt were 5.21± 0.476 mg/kg, 3.76± 0.205 mg/kg, 3.01± 0.135 mg/kg, 1.14± 0.094 mg/kg and 0.82± 0.107 mg/kg respectively.

**Cadmium**: cadmium level was found 0.011–0.277mg/kg in all t-shirt samples.

**Lead**: lead concentrations were found to be 3.40 ± 0.019 mg/kg, 2.71± 0.013 mg/kg, 0.33± 0.018 mg/kg, 0.20± 0.017mg/kg and 0.07± 0.005mg/kg in, red, blue, black, yellow and green-colored t-shirt samples respectively.

**Cobalt**: cobalt concentration ranging from 1.33±0.213 to 3.94±0.021 was found.

## Discussion

Among the studied heavy metals, copper had the highest concentration in green, blue, and black color T-shirts due to ferrocyanide and copper-acetates being used like green, blue, and red-brown dyes and pigments in the textile industry [10, 21]. Similarly, because chromium is

**Table 4. Recovery of heavy metals in T-shirt samples.**

| Analyzed heavy metals | Mean conc. in sample (mg/kg) | Amount of heavy metal added (mg/kg) | Mean conc. found in spiked sample(mg/kg) | % Recovery |
|---|---|---|---|---|
| Cu | 93.37 | 5 | 98.30 | 98.60 |
| Cr | 3.76 | 3 | 6.81 | 101.66 |
| Co | 1.92 | 3 | 4.78 | 95.33 |
| Cd | 0.1 | 5 | 5.02 | 98.40 |
| Pb | 0.33 | 5 | 5.29 | 99.20 |

**Table 5. Concentration of heavy metals (mg/kg) found in a different colored T-shirt by Atomic Absorption spectroscopy.**

| Sample of T-shirt colors | Analyzed heavy metals | | | | |
|---|---|---|---|---|---|
| | Copper(mg/kg) (Mean ± SD) | Chromium mg/kg) (Mean ± SD) | Cobalt (mg/kg) (Mean ± SD) | Cadmium(mg/kg) (Mean ± SD) | Lead(mg/kg) (Mean± SD) |
| Black | 93.37 ±3.192 | 3.76±0.205 | 1.92± 0.161 | 0.017±0.005 | 0.33± 0.018 |
| Red | 53.68 ± 4.548 | 1.14± 0.094 | 3.27± 0.128 | 0.27± 0.007 | 3.40± 0.019 |
| Yellow | 31.53 ± 4.804 | 5.21± 0.476 | 1.68 ± 0.191 | 0.09± 0.002 | 0.20± 0.017 |
| Blue | 177.01± 2.305 | 3.01± 0.135 | 3.94 ± 0.021 | 0.016±0.005 | 2.71± 0.013 |
| Green | 127.16± 6.741 | 0.82± 0.107 | 1.33 ± 0.213 | 0.018±0.002 | 0.07± 0.005 |

SD- Standard deviation

used as a metal complex dye in polyamide black fabrics, the majority of T-shirt samples had high levels of this element [22]. On the other hand, cadmium was found to be within normal levels except in red and yellow T-shirts. However, low levels of cobalt were identified in all T-shirt samples, ranging from 1.33±0.213 to 3.94±0.021mg/kg. Maximum lead concentrations were found to be 3.40 ± 0.019 mg/kg for red-colored samples and 2.71± 0.013 mg/kg for blue-colored samples. The metal concentration in the T-shirts we investigated was also compared to the 100 limit value of mg/kg OEKO Tex standard and with literature in Tables 6 and 7 respectively. In this investigation, the amounts of Pb, Cu, and Cr obtained in red and green colored T-shirt samples were above the OEKO Tex suggested standard value. On the other hand, the amounts of cadmium and cobalt were found to be within the Oeko-Tex guidelines. Heavy metal concentrations in our study were found to be similar the other study [23, 24]. Heavy metal concentrations in the samples investigated could be used to predict skin toxicity and estimate human exposure and health hazards. When metals are present in textile materials above specified levels, they may present a risk to human health through absorption through the skin [25]. The amounts of heavy metals in the T-shirts under investigation differed from one color to the other. The findings of this study demonstrate that Pb, Cu, and Cr concentrations in T-shirts are indicative of a risk factor for human health.

## Conclusion

In this study concentrations of some heavy metals were analyzed in skin contact fading T-shirts printed for a special program at Mettu town using Atomic Absorption Spectroscopy with a microwave digestion method technique for sample preparation. After determining the levels of selected heavy metals high levels of Cu were found in black, green, blue, and red-colored T-shirts ranging from 26.726–179.315mg/kg. *Cr* exceeded the recommended limits in most samples of T-shirts and was mostly in yellow, black, and blue colors. Cd levels were found to be within normal ranges. However, all T-shirt samples had low levels of cobalt,

**Table 6. Oeko-Tex Standard 100 limit values (mg/ kg) [2015].**

| Heavy metals | Concentration (with skin contact) |
|---|---|
| Pb | 1.0 |
| Cd | 0.1 |
| Cr | 2.0 |
| Co | 4.0 |
| Cu | 50.0 |

**Table 7. Literature values of heavy metals (as mg/ kg) in textile.**

| Comparison Table Heavy metals | Other study (Literature) | Present study | Reference |
|---|---|---|---|
| Cu | 0.76-341mg/kg | 26.726–179.315 | [23] |
| Cr | 0-118mg/kg | 0.713–5.686 | |
| Co | 0.0-28mg/kg | 1.117–3.961 | |
| Cd | 0.10–0.41mg/kg | 0.011–0.277 | |
| Pb | 1.23–1.83mg/kg | 0.065–3.419 | [24] |

ranging from 1.33±2.13 to3.94±0.21. Maximum lead concentrations were found to be 3.40 ± 0.19 mg/kg for red-colored samples and 2.71 ± 0.13 mg/kg for blue colored samples. The metal concentrations in the T-shirts investigated were also compared to the OEKO Tex standard 100 limits and the concentrations of Pb, Cu, and Cr in red and green colored T-shirt samples were above the OEKO Tex suggested standard value. These metals may cause allergic reactions or much worse health impacts. For this reason, it is important to recommend that manufacturers and printing houses must ensure that products are safe and do not pose a risk to the users.

## Acknowledgments

The authors would like to express sincere appreciation and gratitude to my families and dear friends for their technical and moral support and encouragement during all times. Next I would like to thank the editor for the constructive comments on improving an early version of this paper.

## Author Contributions

**Conceptualization:** Milkessa Fanta Sima.

**Data curation:** Milkessa Fanta Sima.

**Formal analysis:** Milkessa Fanta Sima.

**Funding acquisition:** Milkessa Fanta Sima.

**Investigation:** Milkessa Fanta Sima.

**Methodology:** Milkessa Fanta Sima.

**Project administration:** Milkessa Fanta Sima.

**Resources:** Milkessa Fanta Sima.

**Software:** Milkessa Fanta Sima.

**Supervision:** Milkessa Fanta Sima.

**Validation:** Milkessa Fanta Sima.

**Visualization:** Milkessa Fanta Sima.

**Writing – original draft:** Milkessa Fanta Sima.

**Writing – review & editing:** Milkessa Fanta Sima.

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
