## [Decision Letter · Decision Letter 0]

22 Jun 2022

PONE-D-22-14909Determination of some heavy metals and their health risk in T-shirts printed for a special program.PLOS ONE

Dear Dr. Sima,

Thank you for submitting your manuscript to PLOS ONE. After careful consideration, we feel that it has merit but does not fully meet PLOS ONE’s publication criteria as it currently stands. Therefore, we invite you to submit a revised version of the manuscript that addresses the points raised during the review process.

We look forward to receiving your revised manuscript.

Kind regards,

MARIA LUISA ASTOLFI, Ph.D.

Academic Editor

PLOS ONE

Journal Requirements:

 [The authors have no affiliation with any organization a direct or indirect financial interest.] 

[The authors declare that there is no conflict of interest regarding the publication of this manuscript.]

Reviewers' comments:

Reviewer's Responses to Questions

**Comments to the Author**

1. Is the manuscript technically sound, and do the data support the conclusions?

Reviewer #1: Yes

Reviewer #2: Partly

2. Has the statistical analysis been performed appropriately and rigorously? 

Reviewer #1: Yes

Reviewer #2: Yes

3. Have the authors made all data underlying the findings in their manuscript fully available?

Reviewer #1: Yes

Reviewer #2: Yes

4. Is the manuscript presented in an intelligible fashion and written in standard English?

Reviewer #1: Yes

Reviewer #2: No

5. Review Comments to the Author

Reviewer #1: - Experimental parts (sample preparation, elemental quantitative analysis) should be referenced.

- Results and Discussion sections should be developed.

- Tables look bad. It must be redrawn.

- Celsius degrees symbols must be written correctly.

Reviewer #2: Please see comments on the attached document but the general comments are:

The paper needs English editing as there are a lot of grammatical errors and typos. The paper is poorly written and that needs to be improved. Novelty is lacking and nothing much has been contributed to scientific knowledge. The paper can be considered for publication after a major revision.

6. PLOS authors have the option to publish the peer review history of their article (what does this mean?). If published, this will include your full peer review and any attached files.

Reviewer #1: **Yes: **Şana Sungur

Reviewer #2: No

---

## [Author Response · Author response to Decision Letter 0]

18 Aug 2022

based on your comments as much as possible I have corrected all comments

---

## [Decision Letter · Decision Letter 1]

1 Sep 2022

PONE-D-22-14909R1Determination of some heavy metals and their health risk in T-shirts printed for a special program.PLOS ONE

Dear Dr. Sima,

Thank you for submitting your manuscript to PLOS ONE. After careful consideration, we feel that it has merit but does not fully meet PLOS ONE’s publication criteria as it currently stands. Therefore, we invite you to submit a revised version of the manuscript that addresses the points raised during the review process.

ACADEMIC EDITOR: The authors improved the manuscript by addressing most of the reviewers' comments. However, some further corrections are required prior acceptance. I ask the authors to respond in detail to Reviewer 2's comments.

We look forward to receiving your revised manuscript.

Kind regards,

MARIA LUISA ASTOLFI, Ph.D.

Academic Editor

PLOS ONE

Journal Requirements:

Reviewers' comments:

Reviewer's Responses to Questions

**Comments to the Author**

1. If the authors have adequately addressed your comments raised in a previous round of review and you feel that this manuscript is now acceptable for publication, you may indicate that here to bypass the “Comments to the Author” section, enter your conflict of interest statement in the “Confidential to Editor” section, and submit your "Accept" recommendation.

Reviewer #1: All comments have been addressed

Reviewer #2: All comments have been addressed

2. Is the manuscript technically sound, and do the data support the conclusions?

Reviewer #1: Yes

Reviewer #2: Partly

3. Has the statistical analysis been performed appropriately and rigorously? 

Reviewer #1: Yes

Reviewer #2: Yes

4. Have the authors made all data underlying the findings in their manuscript fully available?

Reviewer #1: Yes

Reviewer #2: Yes

5. Is the manuscript presented in an intelligible fashion and written in standard English?

Reviewer #1: Yes

Reviewer #2: Yes

6. Review Comments to the Author

Reviewer #1: It is analyzed by considering a current issue. All the corrections requested by the authors have been made. It was quality work.

Reviewer #2: General comments

Most of the previously raised comments were addressed but the novelty is still lacking. There are still some grammatical errors here and there. The paper can be accepted after the following minor corrections.

Comments

1. In the introduction second sentence, change the word ‘peoples’ to people. the ‘s’ should be removed.

2. Third paragraph of the introduction, Chromium VI should be written as Chromium (VI), the brackets are missing.

3. Last sentence of the third paragraph of the introduction, the word ‘especial’ should be ‘special’

4. Under the results section, the first sentence should be completely removed, it has been repeated several times. The reader by now knows that AAS was used.

5. Under ‘method detection limit and limit of quantification’ the space between the words FOR and Blank should be removed.

6. Under the discussion section, the word represent should be present.

7. Authors should avoid putting references in the conclusion since they are concluding their own findings.

8. The referencing style is not the same. For instance, in some references, journal names are written in full whereas others are abbreviated.

7. PLOS authors have the option to publish the peer review history of their article (what does this mean?). If published, this will include your full peer review and any attached files.

Reviewer #1: **Yes: **Şana Sungur

Reviewer #2: No

---

## [Author Response · Author response to Decision Letter 1]

6 Sep 2022

response to reviewer comments attached to files as response to reviewer

---

## [Editor Report · Decision Letter 2]

8 Sep 2022

Determination of some heavy metals and their health risk in T-shirts printed for a special program.

PONE-D-22-14909R2

Dear Dr. Sima,

We’re pleased to inform you that your manuscript has been judged scientifically suitable for publication and will be formally accepted for publication once it meets all outstanding technical requirements.

Kind regards,

MARIA LUISA ASTOLFI, Ph.D.

Academic Editor

PLOS ONE

---

## [Editor Report · Acceptance letter]

11 Sep 2022

PONE-D-22-14909R2 

Determination of some heavy metals and their health risk in T-shirts printed for a special program 

Dear Dr. Sima:

I'm pleased to inform you that your manuscript has been deemed suitable for publication in PLOS ONE. Congratulations! Your manuscript is now with our production department. 

Kind regards, 

on behalf of

Dr. MARIA LUISA ASTOLFI 

Academic Editor

PLOS ONE